# Extreme Ultraviolet Lighting Using Carbon Nanotube-Based Cold Cathode Electron Beam

**DOI:** 10.3390/nano12234134

**Published:** 2022-11-23

**Authors:** Sung Tae Yoo, Kyu Chang Park

**Affiliations:** Department of Information Display, Kyung Hee University, Dongdaemun-gu, Seoul 02447, Republic of Korea

**Keywords:** extreme ultraviolet, field emission, carbon nanotube, electron beam, photolithography

## Abstract

Laser-based plasma studies that apply photons to extreme ultraviolet (EUV) generation are actively being conducted, and studies by direct electron irradiation on Sn for EUV lighting have rarely been attempted. Here, we demonstrate a novel method of EUV generation by irradiating Sn with electrons emitted from a carbon nanotube (CNT)-based cold cathode electron beam (C-beam). Unlike a single laser source, electrons emitted from about 12,700 CNT emitters irradiated the Sn surface to generate EUV and control its intensity. EUV light generated by direct irradiation of electrons was verified using a photodiode equipped with a 150 nm thick Zr filter and patterning of polymethyl methacrylate (PMMA) photoresist. EUV generated with an input power of 6 W is sufficient to react the PMMA with exposure of 30 s. EUV intensity changes according to the anode voltage, current, and electron incident angle. The area reaching the Sn and penetration depth of electrons are easily adjusted. This method could be the cornerstone for advanced lithography for semiconductor fabrication and high-resolution photonics.

## 1. Introduction

Laser-produced tin (Sn) plasma is a commonly used method for producing extreme ultraviolet (EUV) light [1]. Researchers are struggling to improve high-power lasers for this application based on a complicated system using a single laser source [1,2,3]. EUV light with a wavelength of 13.5 nm is generated by the excitation and relaxation of inner electrons of atoms (e.g., Sn^20+^) [1]. Currently, EUV is created by using a laser to form the hot plasma through atomic ionization by photons [3,4]. The hot plasma creates debris of high-energy particles that can damage the optics [1,2,3,4].

Field emission (FE) emits electrons through quantum mechanical tunneling in a high electric field at room temperature and in high or ultra-high vacuum [5]. FE is known for its advantages such as fast switch-on time, compact size, high emission current density and resistance to temperature fluctuations [5,6,7]. The electrons emitted by the FE can be used in a variety of vacuum electronic devices, such as scanning electron microscopy (SEM) for high-resolution imaging [8], microwave amplifiers [9], compact X-ray sources with fast switching [10], flat panel displays [11], and UV light sources [12,13]. To improve the FE properties, the work function of the emitter material should be lowered, and the tip of the emitter sharpened to increase the field enhancement factor. One of the main candidates for FE sources is carbon nanotube (CNT) emitters. They have excellent electrical, mechanical, chemical, and structural properties [6,7]. Because of its extraordinary properties, CNTs have many applications [8,9,10,11,12,14,15,16]. CNTs are synthesized using arc discharge, laser ablation, chemical vapor deposition (CVD), and plasma enhanced chemical vapor deposition (PECVD) [15,16]. For FE enhancement, high-aspect-ratio CNTs were adopted as emitters, and vertically grown CNT emitters were synthesized by PECVD to make sharp tips.

Here, we used CNT emitters as electron source for FE and named them CNT based cold cathode electron beam (C-beam) and applied them to EUV lighting. Our approach is based on EUV light generated by electrons irradiating Sn directly. Electrons from the triode-structured C-beam collide and react with Sn target to generate EUV light. The intensity of EUV light varies depending on the characteristics of the irradiated electrons. This EUV generation and its intensity change were demonstrated by a photodiode equipped with a Zr filter and a polymethyl methacrylate (PMMA) photoresist.

## 2. Materials and Methods

Nickel (Ni) used as a catalyst for fabricating the CNT emitters was deposited using sputtering. The thickness of this thin film is about 30 nm. To pattern the Ni layer defined at the desired location for CNT emitter growth, a traditional photolithography process and Ni etching were performed [17]. CNT emitters were grown onto the Si wafer using PECVD [18]. The positive bias supplied to the mesh electrode and the negative bias injected into the substrate were 300 V and −600 V, respectively. Ammonia (NH_3_) and acetylene (C_2_H_2_) gases were supplied to maintain a total gas pressure of 2.0 Torr during growth. The growth time of the CNT emitter was 120 min. After CNT emitters on Si wafers were grown in a cone shape, they were exposed to hydrofluoric acid for 3 s, and photoresist was spin-coated at 2000 rpm, followed by annealing at 800 °C for 1 h for graphitization [19]. CNT emitters were examined with a scanning electron microscope (SEM; Hitachi S-4700, Tokyo, Japan).

The spectrum of the visible light region generated by the interaction of electrons with the Sn surface was measured using Avanspec-ULS2048 (Avantes, Apeldoorn, the Netherlands). The spectrometer was calibrated to National Institute of Standards and Technology (NIST) traceable standards (200–1099 nm) using a deuterium/halogen light source. This visible light was measured by attaching a spectrometer to a quartz window with a thickness of 10 mm. To measure the intensity of EUV light, a photodiode (SXUV100, OPTO DIODE, Camarillo, CA, USA) was used. In order to remove light having a wavelength of 30 nm or more, a Zr filter (Lebow Co., Philadelphia, PA, USA) of 150 nm was mounted on a photodiode and EUV intensity was measured [20,21]. The current of the photodiode was measured using a Keithley 2400 source meter having a current resolution of 10 pA.

## 3. Results

### 3.1. Carbon Nanotube-Based Cold Cathode Electron Beam for Extreme Ultraviolet Lighting

#### 3.1.1. Schematic

Figure 1 shows the construction for EUV lighting. In the FE structure setup, a C-beam employing CNT emitters was used. The C-beam is composed of a cathode in which the CNT emitters are located, and a gate made of metal mesh that controls the quantity of electrons emitted. The anode made of Sn and the C-beam are separated by a vacuum gap. An SEM image of CNT emitters islands arranged at a constant pitch of 0.5 mm left and right is shown in Figure 1b. Those islands consist of elaborately synthesized CNT emitters with a bottom diameter of 3 μm and a height of 40 μm. The number of CNT emitters used for EUV generation is approximately 12,700. CNT emitters were grown vertically at the desired location by a photolithography process at 15 μm intervals. Conical CNT emitters emit electrons and collide with Sn target by controlling gate voltage which then are injected into the gate and Sn anode. Electron collisions and excitation produce EUV and visible light.

#### 3.1.2. Current-Voltage Characteristics

For field emission, the metal mesh is located 150 μm away from the silicon substrate. The metal mesh is used as a gate to modulate the current emitted from the CNT emitters [22], and the integration of the gate and CNT emitters acts as C-beam. As shown in Figure 2, when the gate voltage applied to the metal mesh is 1.3 kV, the anode current (I_a_), which is equivalent to the total number of electrons reaching the Sn, is 3.1 mA. This means an anode current density of 86.9 mA/cm^2^ at an electric field of 8.7 V/μm. The transmittance of electrons through the gate, which is the ratio of the anode to cathode current, is 89.4%. By applying a voltage to the gate, it is possible to effectively control the emission current from the CNT emitters based on the cold cathode.

### 3.2. EUV Confirmation by Photodiode and Filter

#### 3.2.1. Photocurrent Response to Anode Voltage Change

A 150 nm-thick Zr filter was mounted on a photodiode to measure EUV generation and intensity. I_a_, which is the anode current reaching on the Sn, was fixed at 0.5 mA, and the anode voltage (V_a_) applied to Sn was changed through 5, 10, and 15 kV. Figure 3a illustrates the photocurrent (I_photo_) variation when supply photodiode voltage (V_photo_) from −2 V to 2 V to a photodiode mounted with a Zr filter. When V_photo_ of -1 V is applied to the photodiode, the dark current, which is the current when there is no light, is 5.9 × 10^−10^ A. When the voltage applied to Sn was changed from 5 kV to 15 kV, I_photo_ increased as V_a_ increased, and at 15 kV bias, it reached 5.8 × 10^−6^ A. The change in I_photo_ confirmed EUV generation at higher V_a_. The electron-hole pairs generation increased with higher V_a_. EUV generation depend on the V_a_, because higher V_a_ means stronger electron impact energy on Sn target. At this time, visible light generated by electrons emitted from the C-beam can be observed with the bare eyes. Figure 3b–d are optical images of these lights when V_a_ is 5, 10, and 15 kV, respectively. Figure 3e is a photo of reduced magnification when V_a_ is 15 kV. In the case of 15 kV, the area of light shown in the digital photo is 30.3 mm^2^. As shown in Figure 4, the peak wavelengths were observed in the visible region and the wavelengths were compared with the atomic spectrum of the Sn database of NIST [23]. As V_a_ increases, visible light emission intensity increases, and peaks identified in the visible region are composed of neutral Sn (Sn I), single ionized Sn (Sn II), and double ionized Sn (Sn III) [24]. These excited Sn species are generated by directly irradiating electrons to the Sn target. Electrons emitted from the C-beam create excited Sn atoms and ions from the Sn target due to electron impact energy [25,26,27]. Excitation and ionization of Sn atoms by electron bombardment energy produces EUV and visible light during relaxation process of electrons. From the interaction of electrons with the target, it forms an excited state with a temperature of less than 1 eV in the vicinity of the target surface [25].

#### 3.2.2. Photocurrent Response to Anode Current Change

To explore the effect of I_a_ on the EUV light, a photodiode equipped with a Zr filter was also used at a V_a_ of 15 kV and an incident angle of 40 degrees. V_photo_ was changed from −2 V to 2 V, and I_a_ was changed with the mesh gate bias modulation. As shown in Figure 5, I_photo_ gradually increases from 10 μA to 0.7 mA. The intensity of EUV light could be controlled by modulation of I_a_. As shown in Figure 3, V_a_ affect EUV lighting, and I_a_ also affects EUV lighting. This suggests that increasing the impact power (current times voltage) at the anode could increase the intensity of EUV lighting.

#### 3.2.3. Photocurrent Response According to the Electron Incident Angle

The intensity of the EUV light is related to the penetration depth of electrons into the Sn. Simulations of the penetration depth were performed using the Monte Carlo method with the ‘CASINO’ software program. Figure 6a shows the electron trajectory incident on Sn at 40 degrees and 15 kV. In Figure 6b, the percentages of the contour lines represent the electron energy loss. For example, ‘5%’ indicates that 95% of the electron energy is absorbed by Sn [13]. To investigate the effect of incident angle on penetration depth of electrons, the point at which 95% of the electron energy is absorbed by Sn was defined as the penetration depth. When electrons are incident on Sn at 40 degrees at an anode voltage of 15 kV, the penetration depth is 444 nm, the point at which 95% of the electron energy is absorbed by Sn.

The penetration depth depends on V_a_, and varying V_a_ from 5 kV to 15 kV increases the penetration depth from 93 nm to 444 nm. In this case, the incident angle is fixed at 40 degrees. Figure 7 represents the penetration depths through this analysis. The penetration depth for 15 kV at 60 and 20 degrees was 381 nm and 515 nm, respectively. The penetration depth increased by 1.35 times by the incident angle decreases from 60 to 20 degree at 15 kV anode bias. The penetration depth increases as V_a_ increases and the electron incident angle decreases.

The change in EUV light intensity as a function of the electron incident angle on Sn was investigated. The incident angles were altered to 20, 40, and 60 degrees, and V_a_ was changed while maintaining I_a_ of 0.5 mA. In the photodiode, the change of I_photo_ was confirmed at V_photo_ of −1 V. At the same V_a_, i.e., 15 kV, an I_photo_ of 5.4 × 10^−6^ A was obtained at an incident angle of 60 degrees. At the incident angle of 40 and 20 degrees, I_photo_ is 5.8 × 10^−6^ A and 6.2 × 10^−6^ A, respectively. By varying the incident angle from 60 to 20 degree, the I_photo_ of 8.0 × 10^−7^ A is increased. However, when we change V_a_ from 5 kV to 15 kV, I_photo_ increased 5.4 × 10^−6^ A at the same incident angle. We could obtain a larger photocurrent with higher V_a_; however, 15 kV is possible maximum V_a_ in our system. The change trend of penetration depth according to V_a_ and electron incident angle in Figure 7 is similar to that of I_photo_ one in Figure 8. The penetration depth correlates with the intensity of EUV light.

### 3.3. Extreme Ultraviolet Light Generated by Direct Irradiation of Electrons for Photolithography

To apply EUV light generated by electron irradiation for photo-lithography, PMMA as photoresist and sapphire as photomask were used. PMMA is a main chain scission type photoresist, and EUV light with an energy of at least 9 eV is required to induce a chemical reaction in this photoresist [28,29]. The bandgap energy of sapphire is ~9 eV [30]. EUV light is generated from the C-beam that passes through the Zr filter of thickness 150 nm as shown in Figure 9a, and then only EUV light can reach the sapphire and PMMA [20,21]. Since EUV light does not transmit through sapphire, the PMMA photoresist in the area where the sapphire is present cannot react with EUV. The area where PMMA is exposed to EUV light without sapphire causes a reaction through direct scission from the polymer chain by EUV light and disappears upon the develop process. The EUV exposure was 30 s, V_a_ was 15 kV, and the gate was driven at 1 kHz and 40% duty, so the current reaching the Sn was 0.4 mA. Figure 9b is an optical microscope image (200× magnification) after EUV exposure and develop process. An optical microscope image at 500× magnification is inset in Figure 9b. Like the SEM images in Figure 9c,d, PMMA remains on the left and disappears in the right area. Energy dispersive X-ray analysis detected carbon, a major component of PMMA, in the region where EUV light was blocked by the sapphire. However, silicon, the main component of silicon wafers, does not change significantly. Electrons emitted from the C-beam excite Sn to create an excited state of Sn and ionized Sn, and the resulting EUV light generated by direct irradiation of electrons is sufficient to react with the PMMA.

## 4. Discussion

Since electrons from the C-beam are directly irradiated onto Sn, the intensity of EUV light can be controlled by the anode voltage, current, and angle of incidence of the electrons emitted from the C-beam. In addition, electron emission can be modulated by adjusting the arrangement of the CNT emitters [17], improving the structure of the C-beam [31], and using multiple irradiation through modularization [10]. Since Sn is directly excited by electrons, the area of electrons reaching Sn can be matched with size control of C-beam from the micro-meter [8] to centi-meter scale [31], allowing higher efficiency of EUV lighting with beam size adjustment. By irradiating electrons directly to the solid target, EUV generation could realize debris-free operation with compact size and low-cost without damaging the solid target [32,33]. When electrons emitted by the C-beam are directly irradiated to the Sn target, EUV would be generated while minimizing debris at a point where Sn is not damaged by controlling the intensity of EUV and the area of electrons reaching Sn.

## 5. Conclusions

In conclusion, we performed experiments to confirm EUV light generation based on electrons emitted by field emission (FE) mechanism from CNT emitters. EUV lighting can be observed with the bare eyes in the visible light generation during relaxation of Sn ions, it consists of neutral, single ionized, and doubly ionized Sn. EUV light generated by the impact of electrons with Sn was verified using a PMMA, a photoresist that reacts with light with energies greater than 9 eV, and a photodiode equipped with a 150 nm-thick Zr filter. Since electrons are directly irradiated onto Sn, we observed changes in EUV light with the intensity depending on the anode voltage, current, and electron incident angle. These are related to the penetration depth of electrons into Sn. To increase EUV light intensity, additional studies are needed on the effect of Sn crystallinity, structural improvement of C-beam, and multi-beam irradiation technique. EUV lighting generated by cold cathode electron beam irradiation would accelerate the progress of many scientific and technological fields, especially contributing to advanced lithography and high-resolution imaging technologies.

## Figures and Tables

**Figure 1 nanomaterials-12-04134-f001:**
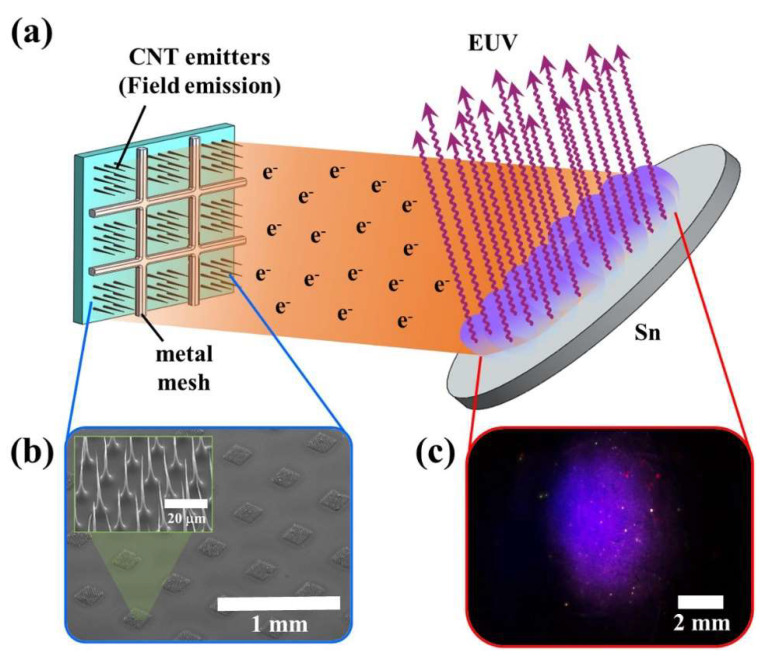
Schematic for extreme ultraviolet lighting with carbon nanotube-based cold cathode electron beam. (**a**) Configuration for extreme ultraviolet generation. (**b**) Scanning electron microscope image of carbon nanotube emitters. (**c**) Photo image of Sn surface.

**Figure 2 nanomaterials-12-04134-f002:**
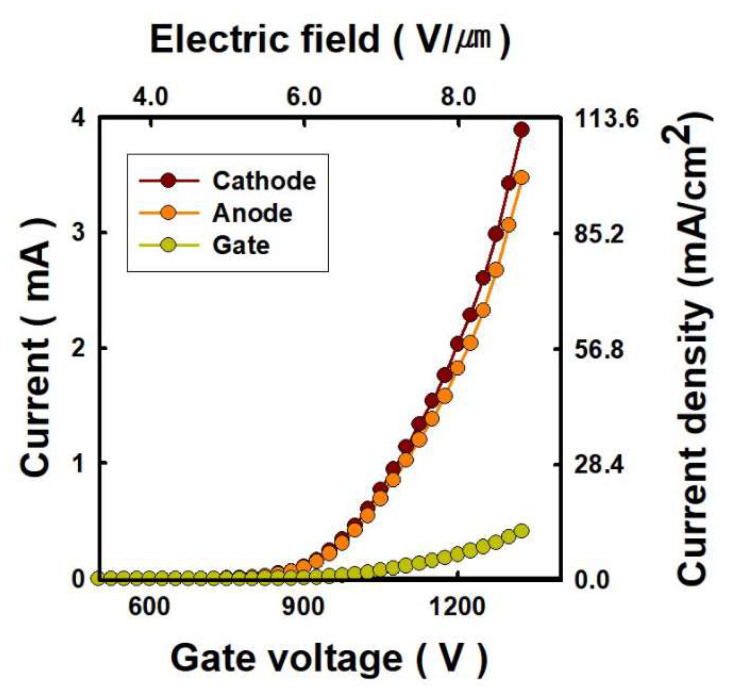
Current-voltage characteristics of carbon nanotube-based cold cathode electron beam. The anode is biased at 10 kV.

**Figure 3 nanomaterials-12-04134-f003:**
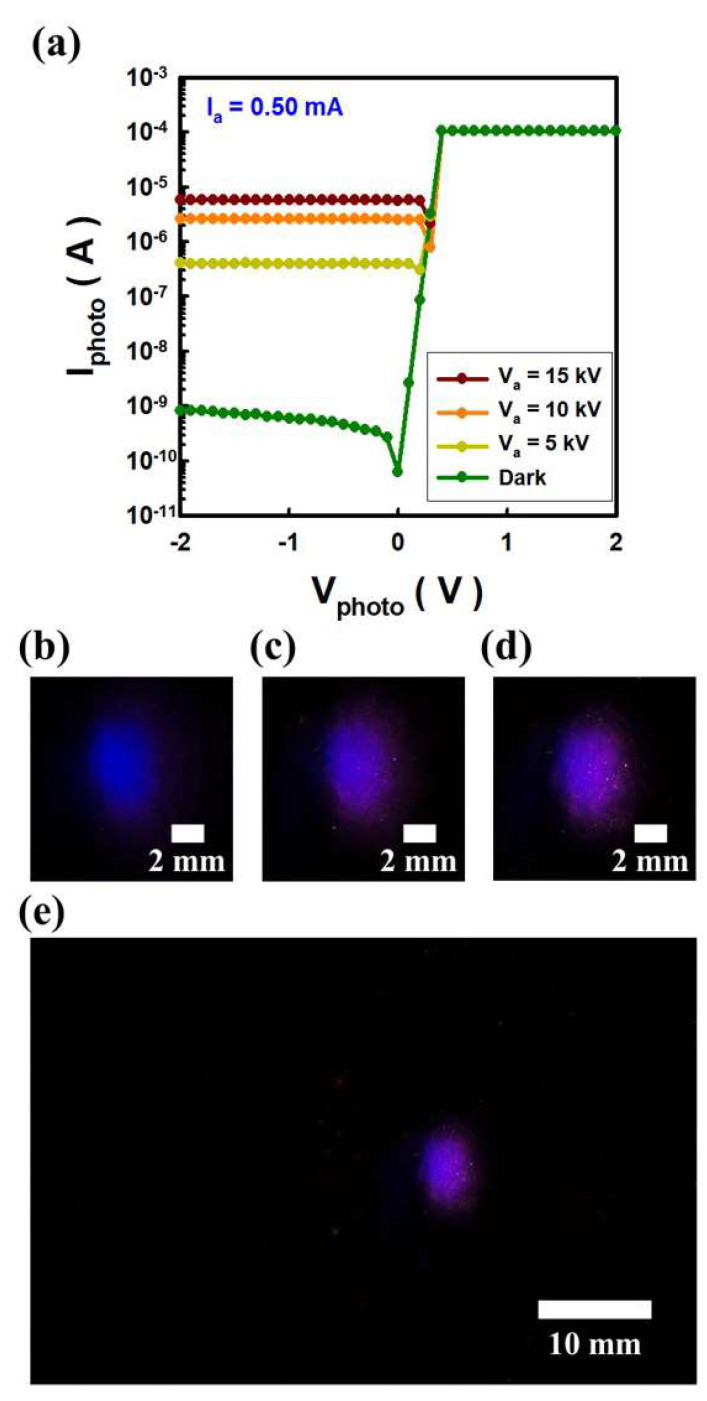
Extreme ultraviolet lighting of Sn target excited by cold cathode electrons irradiation. (**a**) Photocurrent response of EUV lighting through photodiode (SXUV100, OPTO DIODE) measurement. (**b**–**d**) are optical images when the anode current of 0.5 mA is fixed and the anode voltage is 5, 10, and 15 kV, respectively. (**e**) is a reduced-magnification image when the anode voltage is 15 kV.

**Figure 4 nanomaterials-12-04134-f004:**
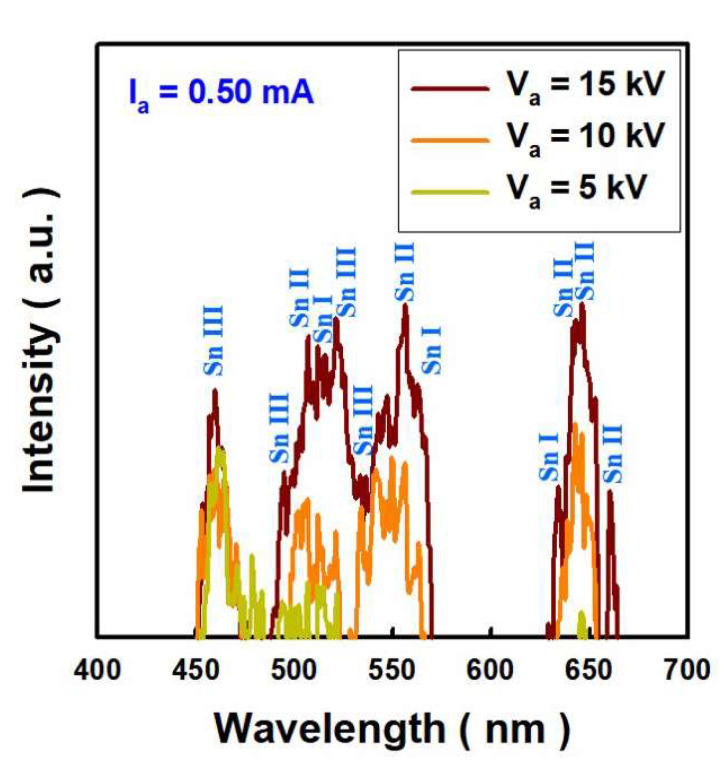
Visible light emission spectra of Sn target excited by cold cathode electrons irradiation. The light generated by electrons in the visible region is composed of neutral Sn (Sn I), single ionized Sn (Sn II), and double ionized Sn (Sn III).

**Figure 5 nanomaterials-12-04134-f005:**
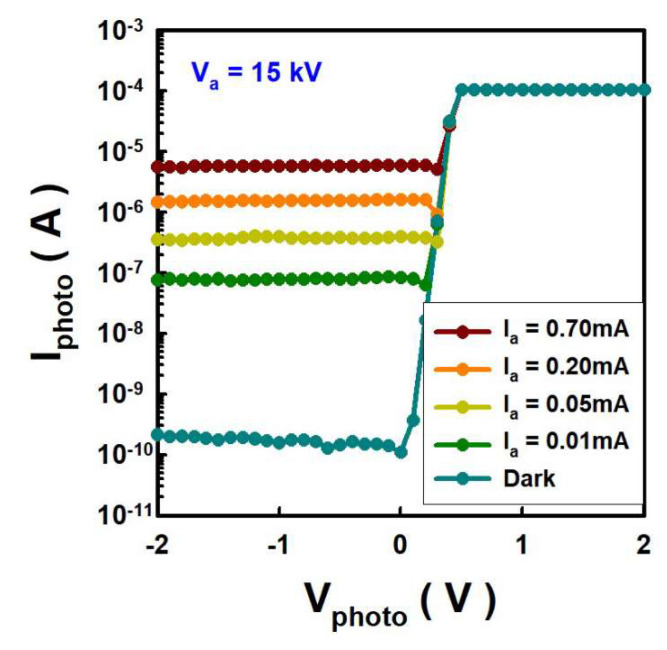
The anode voltage is fixed at 15 kV, and the photocurrent increases as the anode current increases.

**Figure 6 nanomaterials-12-04134-f006:**
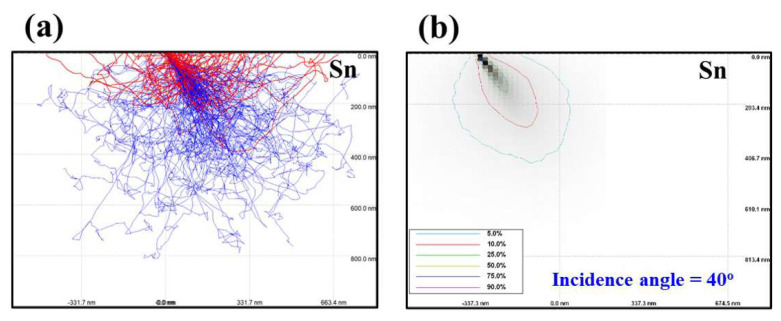
(**a**) Electron trajectory and (**b**) electron energy loss contour when electrons are incident on Sn at an anode voltage of 15 kV at 40 degrees.

**Figure 7 nanomaterials-12-04134-f007:**
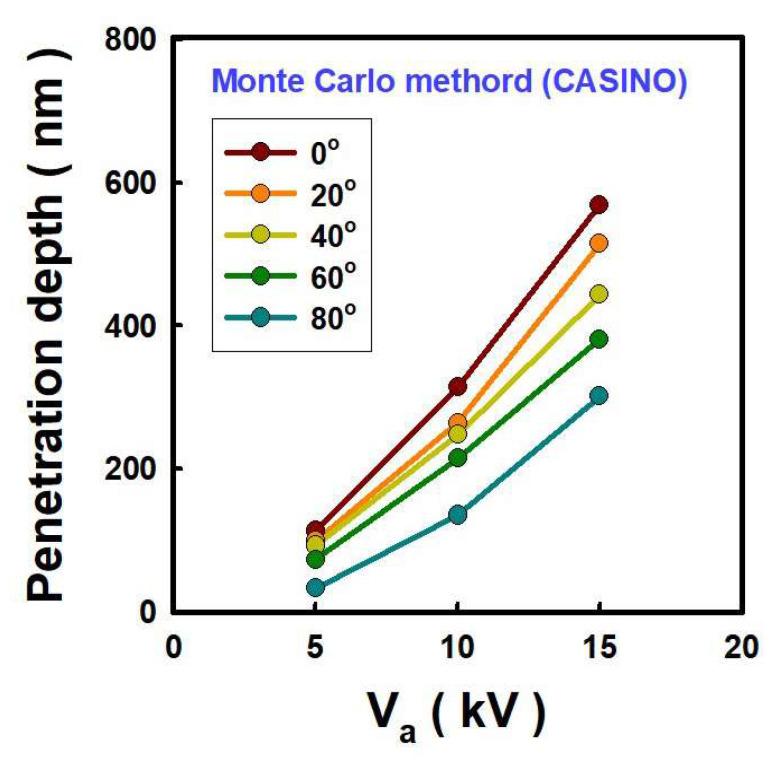
Penetration depth with different anode voltage and electron incident angle injected into Sn.

**Figure 8 nanomaterials-12-04134-f008:**
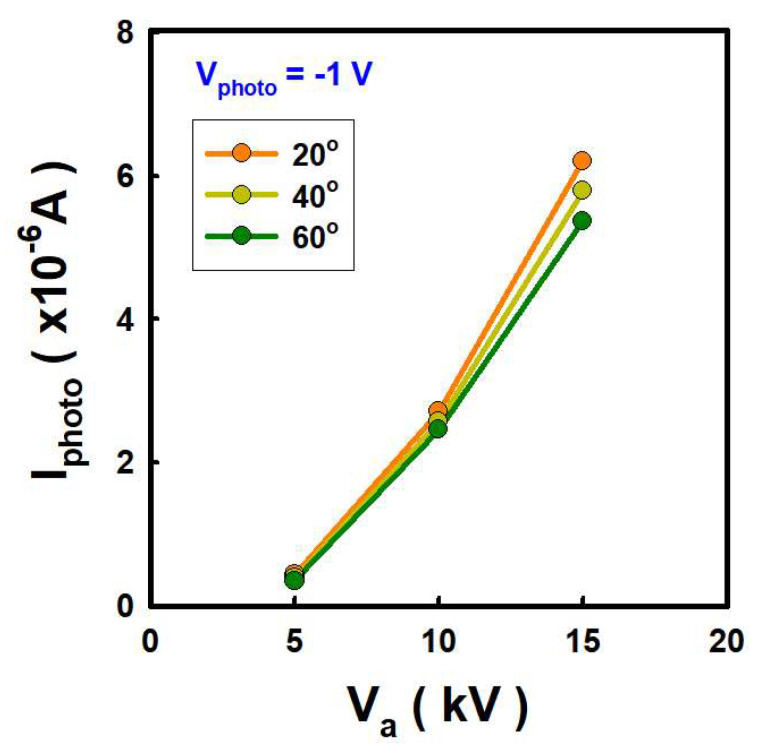
Photocurrent at reverse bias voltage of −1 V with different anode voltage and electron incident angle.

**Figure 9 nanomaterials-12-04134-f009:**
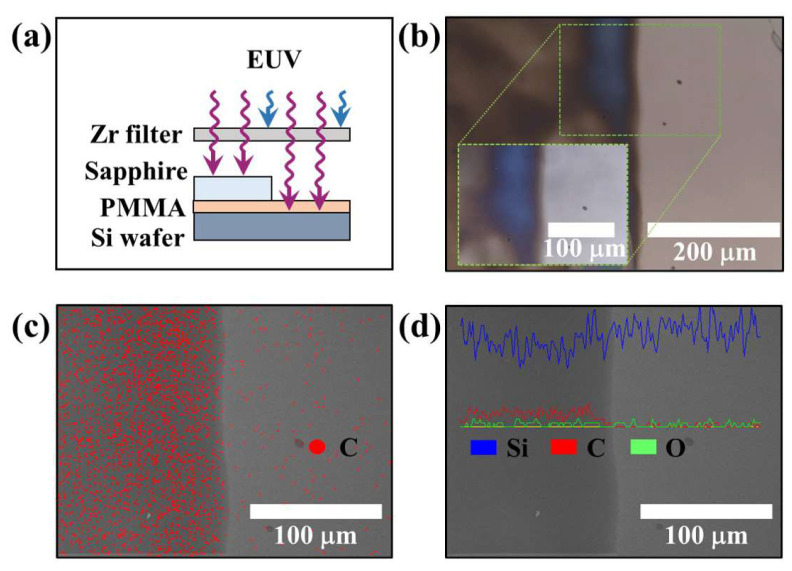
Lithography of polymethyl methacrylate photoresist was performed using extreme ultraviolet light produced through C-beam. (**a**) Schematic view of the extreme ultraviolet light with 150 nm Zr filter. The 150 nm Zr filter only transmits EUV light. Sapphire was used as a photomask, and polymethyl methacrylate was used as a photoresist. (**b**) Optical microscope image after exposure and develop. (**c**) Mapping of carbon elements measured with energy dispersive X-ray analysis. (**d**) Line profile of the element composition measured with energy dispersive X-ray analysis.

## Data Availability

Not applicable.

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
