# Peer review of "Extreme Ultraviolet Lighting Using Carbon Nanotube-Based Cold Cathode Electron Beam"

_nanomaterials, 2022, doi:10.3390/nano12234134_

Round 1

Reviewer 1 Report

A novel method of producing extreme ultraviolet light is demonstrated using carbon nanotube based field emitters.  This has the potential to produce EUV at a lower cost and with a lower chance of generating particles that can damage the system optics. 

 The laser-tin plasma method relies on the tin atoms being in a plasma.  The authors should address the differences resulting from the tin being in a solid phase rather than a plasma.  One would guess that non-radiative processes would dominate in a solid, as compared to a plasma.

This manuscript could be improved by making comparisons to the CNT literature involving electron field emission.  Is what is reported in this paper the same as what has been previously reported? 

Please indicate the crystallinity of the tin layer. Is it polycrystalline? Does it have a preferred orientation?  I would expect its orientation to have an impact on its ability to generate EUV. 

In the paragraph under “Discussion,” the author writes about the limitation of the laser plasma method for generating EUV, specifically debris generation.  However, this is irrelevant as the present study did not address or measure debris generation by this new method.  Effectively, the authors are claiming without providing any evidence that their method does not produce debris, but this is not something they measured.  Therefore, this discussion on debris generation by the laser-plasma method should be deleted. 

While generally well-written, there are some passages that could be combined together to avoid being repetitious.  These are indicated in the accompanying marked manuscript.

Author Response

Pls refer attached point-by-point response.

Reviewer 2 Report

Dear Authors

The manuscript is focused on the novel method of EUV generation by irradiating Sn with electrons emitted from a carbon nanotube (CNT)-based cold cathode electron beam (C-beam).

The following suggestion and comments should be taken:

1. The authors could insert more numerical data into the Abstract for enhancement of the manuscript.

2. The overall English needs to be improved. Please seek guidance from a native English speaker if possible ("the" "a", commas, plural form and others could be corrected).

3. The introduction section needs enhancement 1-3 sentences about carbon nanotubes and their potential applications. Please cite (1) Microsyst Technol 2021. https://doi.org/10.1007/s00542-021-05211-6 (2) Materials 2021, 14(9), 2448; https://doi.org/10.3390/ma14092448 (3) Renew.  Sustain. Energy Rev.2017,68, 234–246

4. Figure 3. Why is only one magnification in b, c and d? Could the authors add other magnifications?

5. Could the authors include the standard deviation of the analysis?

6. Figure 9. Could the authors add other magnifications?

7. Authors are suggested to describe some future plans in conclusions.

Author Response

(The authors gave the same response as above.)

Round 2

Reviewer 2 Report

Dear Authors

I recommend this manuscript for publication.